# Instruments for Patient Safety Assessment: A Scoping Review

**DOI:** 10.3390/healthcare12202075

**Published:** 2024-10-18

**Authors:** Elisabete Nunes, Fernanda Sirtoli, Eliane Lima, Greyce Minarini, Filomena Gaspar, Pedro Lucas, Cândida Primo

**Affiliations:** 1Nursing Research, Innovation, and Development Centre of Lisbon [CIDNUR], Escola Superior de Enfermagem de Lisboa, Avenida Professor Egas Moniz, 1600-190 Lisboa, Portugal; elianelima66@gmail.com (E.L.); mfgaspar@esel.pt (F.G.); prlucas@esel.pt (P.L.); candidaprimo@gmail.com (C.P.); 2Centro de Ciências da Saúde, Campus de Maruípe, Universidade Federal do Espírito Santo, Avenida Marechal Campos, 1.468, Vitória 29047-105, ES, Brazil; fernandacordeirosirtoli@gmail.com (F.S.); greycepoly5@hotmail.com (G.M.)

**Keywords:** safety management, patient safety, healthcare quality, evaluation of research programs and tools, nursing management, review

## Abstract

Background: Patient safety is an important component of healthcare service quality, and there are numerous instruments in the literature that measure patient safety. This scoping reviewaims to map the instruments/scales for assessing patient safety in healthcare services. Method: This scoping review follows the JBI methodology. The protocol was registered on the Open Science Framework. Eligibility criteria were defined based on studies that include instruments or scales for assessing patient safety in healthcare services, in any language, and without temporal restrictions. It adhered to all scoping review checklist items [PRISMA-ScR], with searches in the Embase, Lilacs, MedLine, and Scopus databases, as well as the repository of the Brazilian Digital Library of Theses and Dissertations. Two independent reviewers performed selection and data extraction in July 2023. Results: Of the 4019 potential titles, 63 studies reported on a total of 47 instruments/scales and 71 dimensions for patient safety assessment. The most-described dimensions were teamwork, professional satisfaction, safety climate, communication, and working conditions. Conclusion: The diversity of instruments and dimensions for patient safety assessment characterizes the multidimensionality and scope of patient safety. However, it hinders benchmarking between institutions and healthcare units.

## 1. Introduction

Patient safety is fundamental to healthcare delivery in all settings. However, in the 21st century, adverse events, avoidable errors, and risks associated with healthcare continue to pose significant challenges to patient safety worldwide. Studies indicate that an average of one in ten patients is subject to an adverse event while receiving hospital care in high-income countries. In contrast, in low- and middle-income countries, this estimate is one in four patients, with 134 million adverse events occurring annually because of unsafe care in hospitals, which contributes to about 2.6 million deaths. Overall, 60% of deaths in low- and middle-income countries are due to unsafe and low-quality care [1,2].

Patient safety is a crucial component of healthcare service quality, universally defined as patient care that is free from harm resulting from complications caused by or stemming from that care. While people typically associate patient safety with hospital care, unsafe care is a problem that spans the entire healthcare system, including primary and ambulatory care [3,4].

It is essential to assess real issues related to safe practices in healthcare settings. Patient safety and safety culture are integral components of quality indicators in healthcare services, and identifying strengths and weaknesses aims to guide the institution’s strategic plan for improvement and control of healthcare services offered to patients [5,6].

Numerous instruments scattered in the literature measure patient safety and safety culture, varying considerably in terms of general characteristics, assessed dimensions, conducted psychometrics, and applicability [3,6,7]. This scoping review aims to map instruments/scales for assessing patient safety in healthcare services.

## 2. Materials and Methods

This scoping review follows the JBI methodology. The preferred reporting items for systematic reviews and meta-analyses extension for scoping reviews [PRISMA-ScR] checklist was used to ensure that all relevant aspects of scoping reviews were included [8,9,10].

### 2.1. Protocol and Registration

The protocol was developed and registered on the Open Science Framework [https://osf.io/p329w, accessed on 10 October 2024] before initiating the literature search.

### 2.2. Eligibility Criteria

The research question and eligibility criteria were defined based on the PCC mnemonic suggested by JBI [8]. PCC stands for Population [patients], Concept [instruments/scales for assessing patient safety], and Context [healthcare services]. Thus, the research question was defined as follows: “What are the instruments/scales for assessing patient safety in healthcare services?”.

To address the research question, eligibility criteria were established for study selection following the PCC (Table 1).

This strategy involves analyzing research with varying methodological qualities to identify knowledge gaps and develop a broad understanding of the topic, rather than critically synthesizing the evidence. Our approach aligns with established methodological guidelines for scoping reviews, which advise against excluding studies based on quality [8,9,10].

### 2.3. Search Strategy

Following JBI guidelines, the search strategy occurred in three stages [8]. In the first stage, a limited search was conducted on the PubMed electronic database and the Mesh platform on the topic to identify the most-used descriptors in the literature. The descriptors used for the search were selected with the guidance of a librarian experienced in medical literature research and reviews. In the second stage, the search was conducted in the following databases: MEDLINE, EMBASE, LILACS, and Scopus. Grey literature was also consulted using the repository of the Brazilian Digital Library of Theses and Dissertations [BDTD], provided by the Ministry of Science, Technology, and Innovation of Brazil. In the third stage, the bibliographic references of selected records were analyzed to retrieve potential new records that addressed the research question. The search took place in July 2023.

### 2.4. Study Selection

For the literature search in the review, the Rayyan QCRI^®^ platform [web app Scoping Reviews] was employed. The Rayyan program allows for the removal of duplicates and blinded the two researchers [Sirtoli, FC and Minarini, GPSS], who independently conducted the initial study selection, enabling a comparison of agreements and disagreements. A third researcher [Lima EFA] assessed these.

The results were evaluated and selected for inclusion based on the information provided in the title and abstract. The screening was conducted blindly by two authors simultaneously [Sirtoli, FC and Minarini, GPSS], and disagreements regarding the inclusion of studies were resolved through discussion with a third investigator [Lima EFA]. Subsequently, the selected articles were read in full, and their integration into the final selection was defined.

### 2.5. Data Extraction

For data extraction, an Excel form was developed based on the template provided by JBI [8], containing key information from the sources, such as author, reference, results, and findings relevant to the scope of the review question.

### 2.6. Summarizing and Reporting the Results

The data were synthesized. At this stage, attention was paid to the domains assessed, the number of questions, and the context of the application of the instrument. The results were summarized to present an overview of the patient safety assessment instruments according to the distribution of dimensions/domains.

## 3. Results

In total, 4019 articles were identified, with 1462 in the Scopus database, 1892 in LILACS, 89 in PubMed/MEDLINE, 575 in EMBASE, 01 in the Brazilian Digital Library of Theses and Dissertations [BDTD], and 1014 duplicated articles. As recommended by JBI, the PRISMA-ScR flowchart model was used [10]. Figure 1 provides a detailed overview of the study selection process.

The sample consisted of 63 articles, with publications starting in 2007, and most occurring between 2017 and 2021. English was the predominant language of the publications. Of the studies, 19 instruments were applied in research settings in the United States of America, and five were applied in Canada; both countries in North America. Fifteen studies were conducted in other countries in Europe, seven in countries belonging to Oceania, six in the Asian continent, six in the Middle East, and five in countries across Asia and South America, such as Brazil.

Forty-seven instruments [3,4,5,6,7,8,9,10,11,12,13,14,15,16,17,18,19,20,21,22,23,24,25,26,27,28,29,30,31,32,33,34,35,36,37,38,39,40,41,42,43,44,45,46,47,48,49,50,51,52,53,54,55,56,57,58,59,60,61,62,63,64,65,66,67,68,69,70,71,72] were identified, with 27 applicable to intra-hospital care, covering emergency services, and the others were tailored for intra-hospital emergencies [*n* = 4], pre-hospital care by surface or air [*n* = 4], family members in the emergency department [*n* = 1], primary health care [*n* = 4], and any health area [*n* = 7]. The majority of these studies focus on emergency services.

Most articles used instruments to assess the patient safety culture, while others assessed patient safety by focusing on a specific aspect, with closed-ended questions predominating. The Likert scale was commonly used to measure agreement levels, ranging from three to five points. In each instrument, the number of questions varied from 3 to 67, incorporated into 71 dimensions. The instrument Safety Attitude Questionnaire [SAQ] [*n* = 22] and the Hospital Survey on Patient Safety Culture [HSOPSC] [*n* = 17] were frequently used, as presented in Table 2, while Table 3 describes the distribution of patient safety assessment instruments according to dimensions/domains.

## 4. Discussion

Regarding the composition of dimensions and questions, a variety of dimensions were identified in the instruments. Those with the highest number of dimensions were the Hospital Survey on Patient Safety, with 14 dimensions, and those with the highest number of instruments were the Alberta Registered Nurse Survey, Medical Office Survey on Patient Safety Culture, Patient Measure of Safety, and Patient Safety Climate in Healthcare Organizations [PSCHO], with 12 dimensions each, respectively.

The Swedish version of the Hospital Survey on Patient Safety included 51 items covering 14 dimensions of patient safety culture and three single-outcome questions with a 5-point Likert scale [35].

Different instruments comprised 12 dimensions, including the Alberta Registered Nurse Survey, which was applied to nurses working in various intra-hospital sectors in Canada, including emergency services, with responses given on a Likert scale, yes or no, or multiple choice questions [67]. The Medical Office Survey on Patient Safety Culture has 54 questions, with 6 specific to primary care among the 12 domains. It is available in English and Spanish and has a reliability of 0.77–0.90 [42].

The Patient Measure of Safety is another example of a tool that encompasses 12 domains, including communication, individual factors, physical environment, bed management, staff management/workload, dignity and respect, training and education, lines of responsibility, equipment and supplies, supervision and leadership, team factors, and support from central functions [22].

The instrument Patient Safety Climate in Healthcare Organizations has 45 items, using a 5-point Likert scale ranging from “strongly agree” to “strongly disagree.” Based on the principal component analysis, the research evaluates 12 dimensions reflecting general safety climate components, such as top management involvement in patient safety, the existence of a blame culture, and responsiveness of the unit manager to identified safety issues. PSCHO used the High Reliability Organizations Theory [HROT] to guide its tool development process [7,16]. A safety climate survey containing 15 to 20 items based on the PSCHO instrument was applied to managers, physicians, and healthcare professionals in hospital settings, including organizational, unit-based, and interpersonal domains [72].

In the instruments, there was a predominance of dimensions such as teamwork, job satisfaction, safety climate, communication, and working conditions. Some dimensions were less prominent, manifesting in a single instrument with specific characteristics, such as “burnout” in the OCSFS—Organizational Climate Safety Factors instrument (Table 3).

Forty-seven instruments were identified, of which twenty-seven apply for intra-hospital care, covering emergency services, and the remaining instruments were created for intra-hospital emergency [*n* = 4], surface or air pre-hospital care [*n* = 4], family members in emergency services [*n* = 1], primary healthcare [*n* = 4], and any healthcare area [*n* = 7].

Focused on intra-hospital care, covering emergency services, the Manchester Patient Safety Framework [MaPSaF] was developed to support healthcare teams and organizations in developing a safety culture in emergency services [50].

MaPSaF has ten dimensions and aims to encourage proactive behavior, increase awareness of patient safety, identify improvements, assess interventions, and monitor changes [42]. The dimensions include continuous improvement; priority given to safety; system errors and individual responsibility; incident recording; incident assessment; learning and making changes; communication; staff management; staff education; and teamwork [50].

Another tool, the Patient Measure of Safety [PMOS], used in three articles in this review, provides a systematic way to assess patient safety. It is considered a tool applied to healthcare professionals and patients capable of proactively identifying evidence-based contributing factors to safe patient care and signaling areas for hospitals to direct improvements [22,32,59]. PMOS is a 44-item questionnaire with nine domains [27].

Regarding the assessment of patient safety culture in intra-hospital emergency services, this review identified the Revised Professional Practice Environment [RPPE] instrument. The applicability of RPPE involved physicians and nurses to evaluate professionals’ perceptions of their work environment and professional practice. The scale was designed with 39 questions distributed across 8 dimensions, with responses scored on a 5-point Likert scale [67].

Studies using the Safety Attitudes Questionnaire [SAQ] in healthcare settings were identified based on two conceptual models: Vincent’s model for risk and safety analysis and Donabedian’s conceptual model for quality assessment [5,7]. This instrument underwent variations and adaptations to fit the specific reality of each country, setting, and study objects. Therefore, depending on the country where it was applied, this scale ranged from 30 to 60 questions, and responses to each question followed a Likert-type scale of 5 points for the degree of agreement [7,12,13,15,24,26,28,30,38,39,40,42,44,48,51,52,54,63,65,66,67,70,71].

The Health Professional Education in Patient Safety Survey [H-PEPSS] focuses mainly on the socio-cultural aspects of patient safety, including culture, teamwork, communication, managing risk, and understanding human factors. The study results indicate that H-PEPSS is an instrument that is capable of measuring knowledge, skills, and attitudes in patient safety and is largely useful for examining the impact of specific patient safety curriculum initiatives [21].

There is also the Hand Hygiene Self-Assessment Framework [HHSAF], which is a systematic self-assessment tool from the World Health Organization aimed at obtaining a situational analysis of hand hygiene promotion and practices in healthcare facilities [49].

Another assessment tool supporting the implementation of World Health Organization guidelines is the Infection Prevention and Control Assessment Framework [IPCAF], published in 2018. It focuses on key components of effective infection prevention and control programs in acute healthcare, aiming to identify relevant problems or deficiencies requiring improvement through regular form reapplication to document progress over time and identify strengths and gaps that inform future policy [49].

Addressing the family context of the emergency service, the Person-Centred Climate Questionnaire—Family [PCQ-F] consists of 17 items in three dimensions of the psychosocial climate. It aims to analyze aspects of the safety climate of the intra-hospital emergency service perceived by patients’ families and evaluate the centrality of the person in the climate, as perceived by family members. The tool can guide health managers, workers, and stakeholders in the analysis and intervention of the psychosocial climate in long-term care institutions [67].

Among the forty-one instruments identified in the research, three were specifically constructed for primary care [SCOPE, PC-Safe Quest, and EPA]. The Short Child Occupational Profile [SCOPE] is the Dutch version of the Hospital Survey on Patient Safety Culture [HSOPSC] from the US Agency for Healthcare Research and Quality [AHRQ], validated and adapted for use in primary care in the Netherlands [42].

The Dutch HSOPSC consists of 56 questions assessing 11 dimensions of patient safety culture: teamwork across hospital units, teamwork within units, hospital handoffs and transitions, event notification frequency, non-punitive response to error, communication openness, feedback and communication about errors, managerial actions promoting patient safety, hospital management support for patient safety, adequate staff, and general safety perceptions [14,67].

Concerning the PC-Safe Quest, Safe Quest Safety Climate Survey, it is an instrument intended for all members of the primary health care team based on the practice of professionals in the community, containing 30 items grouped into five dimensions: communication, leadership, teamwork, safety systems, and workload [41,42,52].

The European Practical Assessment [EPA] was built from an observational study in nine European countries with diverse healthcare systems as a viable and valid educational technology to measure the organization and management of primary care practices within or between countries or to seek trends over time. The instrument can also provide personalized feedback, including minimum standards [summative], benchmark parameters, and suggestions for practice improvement [formative], either at the initiative of the practice itself or as a part of accreditation. In this context, each country can choose to use the EPA for a summative or formative assessment or a combination of both [42].

This questionnaire [EPA] has 45 items related to patient safety management divided into ten patient safety domains that assess users’ and professionals’ perceptions of the institution’s infrastructure, quality, and safety [42].

The instruments identified in this review contribute to monitoring patient safety in healthcare services, which is essential in the management work process, as the adoption of safe practices directly influences the quality of care provided, the work process of the healthcare team, and the financial costs related to care.

This review is valuable for healthcare management, as managers can select instruments that best fit their contexts and objectives for improving patient safety.

The article selection indicates a focus on organizational and cultural factors (teamwork, communication, leadership, etc.) that influence the safety of care. However, to assess the validity of these tools, outcome measures are essential. Indicators such as reductions in adverse events, healthcare-associated infections, mortality rates, and litigations would provide a clearer understanding of the tools’ effectiveness [11,16]. For example, the study on the WHO surgical checklist is particularly noteworthy, as it evaluates outcomes (complications and mortality) based on the use or non-use of the checklist [73].

Another critical issue is the assessment of care safety based on the skills and competencies of healthcare professionals. A strong safety culture, effective communication, and teamwork are important, but these factors alone are insufficient without understanding how professionals (clinicians, surgeons, nurses) develop their competencies. In other words, the accreditation of organizational quality cannot be separated from the assessment of professional quality [21,54,74].

Our study has some limitations. The scoping review methodology does not imply quality assessment; therefore, our review did not analyze the quality of the included studies. There is a risk of selection and publication bias due to the wide variety of methodological approaches and instruments related to patient safety, and our search strategy may not have fully covered this. Additionally, our search strategy used only one base for the gray literature research [BDTD]. We acknowledge that more databases could have been searched, including those cataloging gray literature. Another limitation was the difficulty in capturing three studies with restricted access. In addition, most of the selected articles did not include information on the effectiveness and validity of these tools.

In this study, we employed a comprehensive search strategy conducted by an experienced librarian. The screening process was performed blinded, and we used a detailed form to summarize the included studies. We also rigorously applied the PRISMA-ScR checklist for a structured evaluation in this scoping review.

## 5. Conclusions

This study identified 47 instruments/scales to assess patient safety in healthcare services. Within these instruments, 71 dimensions of patient safety assessment were identified. The instruments with the highest number of dimensions were the Hospital Survey on Patient Safety, Alberta Registered Nurse Survey, Medical Office Survey on Patient Safety Culture, Patient Measure of Safety, Patient Safety Climate in Healthcare Organizations, and Team Strategies and Tools to Enhance Performance and Patient Safety. The predominant dimensions across the instruments were teamwork, professional satisfaction, safety climate, communication, and working conditions.

This review allowed for the identification of instruments that are adapted and validated for each culture, context, and language. The diversity of patient safety assessment instruments with various dimensions found well-characterizes the multidimensionality and scope of patient safety. However, it poses some challenges for benchmarking between institutions/healthcare units.

## Figures and Tables

**Figure 1 healthcare-12-02075-f001:**
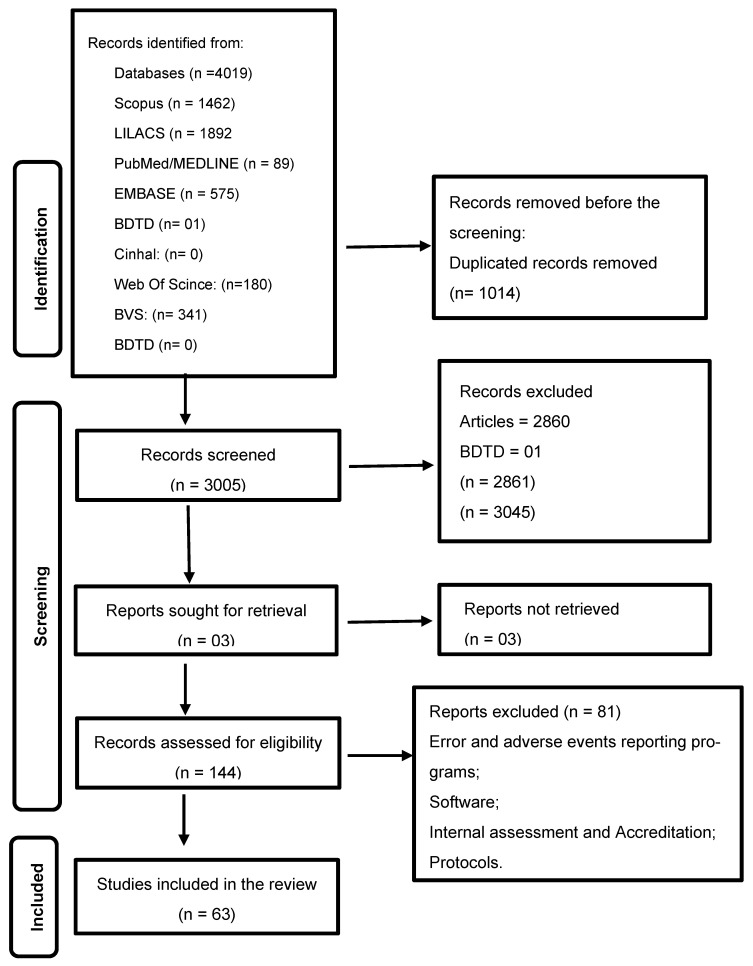
Study selection flowchart—Prisma-SCR. Source: Adapted from Peters et al. [2020] [10].

**Table 1 healthcare-12-02075-t001:** Eligibility criteria for the scope review studies—Brazil, 2023.

Inclusion Criteria *	
Population	Patients
Concept	Concept instruments/scales for assessing patient safety
Context	Healthcare services
Search sources	MEDLINE [OVID], EMBASE, LILACS, Scopus, Brazilian Digital Library of Theses and Dissertations [BDTD]
Study design	Studies that include instruments or scales for assessing patient safety in healthcare services.
Period	No time restriction
Language	All
Availability	Full text available
Descriptors	Safety Management; Patient Safety; Total Quality Management; Process Assessment, Health Care; Evaluation of Research Programs and Tools; Surveys and Questionnaires
Descriptor crossings	Process Assessment, Health Care AND Patient Safety AND Safety Management Safety Management AND Patient Safety AND Evaluation of Research Programs and Tools Patient Safety AND Evaluation of Research Programs and Tools Total Quality Management AND Evaluation of Research Programs and Tools AND Patient safety Process Assessment And Patient Safety AND Health Care AND Evaluation of Research Programs and Tools Surveys and Questionnaires AND Evaluation of Research Programs and Tools
Study in the format of a scientific article, guideline, doctoral thesis, master’s dissertation, or complete abstract published in proceedings or scientific journals.	
**Exclusion Criteria ****	
Studies that do not distinguish instruments/scales for assessing patient safety in healthcare services. Studies that include instruments or scales for investigating adverse events.	
Studies with restricted access and whose request for availability to the authors was not met.	

Source: Authors [2023]. * For study selection, all inclusion criteria must be present. ** The study will be excluded if at least one of the exclusion criteria is present.

**Table 2 healthcare-12-02075-t002:** Distribution of studies according to patient safety assessment instruments by country, number of questions, dimensions, and context of application—Vitória, ES, Brazil, 2023.

	Instrument	Country	Nº of Questions	Nº of Dimensions	Context
1	RPPE [6,67]	Chipre	39	8	α
2	ED Survey Colorado [67]	USA	55	9	β
3	ED Survey Indianapolis [67]	USA	67	10	β
4	Safety beliefs and practices conducted by the Air and Surface Transport Nurses Association [67]	USA	15	4	Ω
5	EMS Safety Climate Scale [67]	USA	20	6	Ω
6	EMS-SI [67]	USA	44	6	Ω
7	Alberta Registered Nurse Survey [67]	Canada	20	12	α
8	Institute for Healthcare Improvement [67]	USA	19	4	β
9	EMS-SAQ [67]	USA	30	6	Ω
10	PCQ-F [67]	Sweden	17	3	δ
11	SAQ [5,7,12,13,15,24,26,28,30,38,39,40,42,44,48,51,52,54,63,65,66,67,70,71]	USA [5], Italy [15], UK [7,65]	41	5	ỵ
		Brazil [6,28,42,66,67]	41	6	α
		Ghana [51], USA [52,63]	56	6	α
		Egypt [42]	60	NR	α
		USA [12,13,39,40,52,54]	30	6	α
		China [24] Taiwan [18,71] Palestine [30] Denmark [70]	32	6	α
		Korea [26]	35	7	α
		Saudi Arabia [38], Australia [44]	36	6	α
		USA [48]	57	7	α
12	SAQ-OR [31]	Portugal	59	6	α
13	SAQ-AV [37,45,57]	The Netherlands [37] Norway [45] Italia [57]	62	5	ỵ
14	TSCS [42]	UK	27	3	⥀
15	EPA [42]	Netherlands	45	5	ỵ
16	MaPSaF [42,50]	New Zealand [42] UK [50]	NR NR	1 9	⥀ α
17	PC-SafeQuest [41,42,52,54]	UK [41], Scotland [42], USA [52,54]	30	4	α
18	MOSPSC [3,42]	USA [42]	54	12	α
		Yemen [3]	44	12	ỵ
19	HSOPSC [6,14,17,23,24,25,33,34,35,42,46,53,58,60,61,68]	The Netherlands [14]	56	11	ỵ
		Portugal [67], Korea [67], South Korea [67], Saudi Arabia [67], Sweden [67], Spain [23,34] The Netherlands [25], USA [25], Taiwan [25] Japan [46]	42	12	α
		Kuwait [43]	22	8	α
		Turkey, Spain [42], China, Iran [6,42]	42	12	β
		USA [33]	39	6	α
		Sweden [35]	51	14	α
		Belgium [17]	42	12	α
		China [24]	29	10	α
		China [53,68]	42	12	β
		Tunisia [60,61]	45	10	β
		Bulgaria [58]	37	11	β
20	AACN HWEAT e HSOPSC [67]	USA	20	6	α
21	SCOPE [42]	The Netherlands	46	3	ỵ
22	PMOS [22,27,32,59]	Australia [32,59]	43	9	α
		UK [22]	42	12	α
		UK [27]	44	9	α
23	NOTECHS [64]	Canada	4	NR	α
24	TEAMS [64]	Canada	11	3	α
25	SEIPS too [64]	Canada	6	NR	α
30	PSCHO [7,16,72]	UK [7], USA [16,72]	45	12	α
31	SOS [7]	UK	NR	NR	α
32	Can-PSC [7]	UK	NR	NR	α
33	OCSFS [11]	USA	30	6	α
34	MSSAPS [47]	France	28	5	α
35	Trigger Tool [41,56]	UK [41] The Netherlands [56]	NR	NR	α
36	IPCAF [49]	Switzerland	NR	5	α
37	HHSAF [49]	Switzerland	NR	5	α
38	HCAHPS [50]	UK	32	8	α
39	Health Professional Education in Patient Safety Survey [H-PEPSS] [21]	Canada	23	6	α
40	WHO’s Surgical Safety Checklist [49]	Sweden	NR	NR	α
41	Patient Safety Culture Questionnaire in Acute Geriatric Units [19]	Germany	7	7	ỵ
42	Specific Questionnaire on Patient Safety in the Laboratory [20]	Spain	62	6	⥀
43	NHSOPSC [29]	Norway	43	12	ỵ
44	ASCN [36]	Iran	32	4	⥀
45	Patient Participation Questionnaire [PPQ] [55]	Denmark	17	4	α
46	Medical Office Survey on Patient Safety Culture [SOPS] [62]	USA	38	10	α
47	Influences on Patient Safety Behaviors Questionnaire [IPSBQ] [69]	Australia	NR	11	α

Legend: TeamS—Team Strategies and Tools to Enhance Performance and Patient Safety; SAQ—Safety Attitudes Questionnaire; SAQ-AV—Safety Attitudes Questionnaire—Ambulatory Version; SAQ-OR—Safety Attitudes Questionnaire/Operating Room; SAQ-EMS—Safety Attitudes Questionnaire—Emergency Medical Service; RPPE—Revised Professional Practice Environment; PSCS—Patient Safety Culture Survey; ED—Emergency Department; EMS—Emergency Medical Services; EMS-SI—Emergency Medical Services—Safety Inventory; PCQ-F—Person-centered Climate Questionnaire—Family; PMOS—Patient Measure of Safety; EPA—European Practice Assessment; TSCS—Teamwork and Safety Climate Survey; MaPSaF—Manchester Patient Safety Framework; PSCHO—Patient Safety Climate in Healthcare Organizations; MOSPSC—Medical Office Survey on Patient Safety Culture; HSOPSC—Hospital Survey on Patient Safety Culture; SCOPE—Safety Culture Questionnaire for General Practice; QSEN-SES- Quality and Safety Education for Nurses Student Evaluation Survey; PC-SafeQuest—Primary Care Safety Questionnaire; OCSFS—Organizational Climate Safety Factors; HHSAF—Hand Hygiene Self-Assessment Framework by WHO; HCAHPS—Hospital Consumer Assessment of Healthcare Providers and Systems; H-PEPSS—Health Professional Education in Patient Safety Survey; NHSOPSC—Nursing Home Survey on Patient Safety Culture; PPQ—Patient Participation Questionnaire; SOPS—Medical Office Survey on Patient Safety Culture; IPSBQ—Influences on Patient Safety Behaviors Questionnaire; TeamS—Team Strategies and Tools to Enhance Performance and Patient Safety; IPCAF—Infection Prevention and Control Assessment Framework by WHO; ASCN—Instrument for the Assessment of Safe Nursing Care; AACN HWEAT—American Association of Critical-Care Nurses Healthy Work Environment Assessment Tool; MSSAPS—Medical Student Safety Attitudes and Professionalism Survey; Can-PSC—Canadian Patient Safety Climate Scale, SOS—Safety Organizing Scale; NOTECHS—Operating Theatre Team Non-Technical Skills Assessment Tool; SEIPS—Systems Engineering Initiative for Patient Safety; NR—Not Reported; α—intra-hospital encompassing emergency service; β—exclusive for intra-hospital emergency; Ω—pre-hospital care, both by surface and/or air; δ—family members in the emergency service; ỵ—primary health care; ⥀—any health care area.

**Table 3 healthcare-12-02075-t003:** Distribution of patient safety assessment instruments according to dimensions/domains—Vitória, ES, Brazil, 2023.

	Dimensions/Domains	Instruments
1	Expectations and actions of unit/service leadership/supervision that favor safety	HSOPSC; PSCS, SAQ, NHSOPSC
2	Teamwork	SAQ, EPA, TSCS, PC-SafeQuest, MaPSaF, MOSPSC, SCOPE, PMOS, HSOPSC; PSCS; SAQ-EMS; RPPE; ED; EMS-SI, MSSAPS, PSCHO, QSEN-SES, SAQ-OR, NHSOPSC, ASCN, SAQ-AV
3	Training	PMOS, EMS, PSCHO, NHSOPSC
4	Team structure	PMOS, NHSOPSC
5	Communication	SAQ, EPA, TSCS, PC-SafeQuest, MOSPSC, ESCOPO, PMOS, HSOPSC; PSCS, RPPE, Institute for Healthcare Improvement, PMOS, PSCHO, Safety Culture Survey, HCAHPS, Patient Safety Culture Questionnaire in Acute Geriatric Units, Specific questionnaire on patient safety in the laboratory, NHSOPSC
6	Leadership	RPPE, PMOS
7	Situation monitoring	ED, Safety Culture Survey
8	Safety perceptions	HSOPSC; PSCS; SAQ-EMS; SAQ; EMS-SI, PSCHO, Safety Culture Survey, NHSOPSC
9	Safety reports [including incident and near-miss notification]	Safety beliefs and practices conducted by the Air and Surface Transport Nurses Association, MaPSaF
10	Frequency of notified adverse events	HSOPSC; PSCS, Alberta Registered Nurse Survey
11	Organizational learning—continuous improvement	PMOS, HSOPSC; PSCS, Safety Culture Survey, PatientSafety Culture Questionnaire in AcuteGeriatricUnits
12	Feedback and communication about errors	MSSAPS, MaPSaF, Safety Culture Survey, HCAHPS, NHSOPSC
13	Non-punitive response to errors	HSOPSC; PSCS, Safety Culture Survey, NHSOPSC
14	Staffing levels	HSOPSC; PSCS, ED
15	Hospital management support for patient safety	HSOPSC; PSCS, Safety Culture Survey, NHSOPSC
16	Management perception	SAQ, EPA, TSCS, PC-SafeQuest, MOSPSC, SCOPE NHSOPSC, SAQ-AV
17	Stress/workload perception	SAQ, EPA, PC-SafeQuest
18	Job satisfaction	SAQ, EPA, SAQ-EMS; SAQ; EMS-SI; Alberta Registered Nurse Survey, OCSFS, PSCHO, SAQ-AV
19	Access to resources	PMOS
20	Equipment design and operation	PMOS
21	Roles and responsibilities	PMOS
22	Information flow	PMOS
23	Ward type and layout	PMOS
24	Issues in handoffs and transitions between units/services	HSOPSC; PSCS
25	Safety climate	SAQ-EMS; SAQ; PCQ-F; EMS-SI; Institute for Healthcare Improvement, The Safety Attitudes Questionnaire, OCSFS, MSSAPS, PSCHO, MaPSaF, HCAHPS, Specific questionnaire on patient safety in the laboratory, SAQ-OR, SAQ-AV
26	Stress	SAQ-EMS; SAQ; EMS-SI, Patient Safety Culture Questionnaire in Acute Geriatric Units, SAQ-OR, SAQ-AV
27	Working conditions	SAQ-EMS; SAQ; EMS-SI, The Safety Attitudes Questionnaire, OCSFS, PSCHO, Patient Safety Culture Questionnaire in Acute Geriatric Units, Specific Questionnaire on Patient Safety in the Laboratory, SAQ-OR, SAQ-AV
28	True collaboration	AACN HWEAT
29	Skilled communication	AACN HWEAT
30	Effective decision-making	AACN HWEAT
31	Meaningful recognition	AACN HWEAT
32	Authentic leadership	AACN HWEAT
33	Climate of everydayness	PCQ-F
34	Climate of hospitality	PCQ-F
35	Internal work motivation	RPPE
36	Control over practice	RPPE
37	Staff relationships with physicians	RPPE, MSSAPS
38	Physical environment	ED, PMOS
39	Nursing	ED
40	Culture	ED
41	Coordination	ED, Patient Safety Culture Questionnaire in Acute Geriatric Units
42	Availability of personal protective equipment	EMS, PMOS
43	Management support	EMS
44	Absence of job hindrances	EMS
45	Cleanliness of workspace	EMS
46	Minimal conflict/good communication	EMS
47	Crowding	Institute for Healthcare Improvement
48	Medication safety	Institute for Healthcare Improvement
49	Quality of care	Alberta Registered Nurse Survey
50	Adverse patientevents	Alberta Registered Nurse Survey, Patient Safety Culture Questionnaire in Acute Geriatric Units
51	Full-time/part-time work	Alberta Registered Nurse Survey
52	Salary	Alberta Registered Nurse Survey
53	Continuing education	Alberta Registered Nurse Survey, MAPSAF, Specific Questionnaire on Patient Safety in the Laboratory
54	Quality assurance program	Alberta Registered Nurse Survey
55	Preceptorship	Alberta Registered Nurse Survey
56	Autonomy	Alberta Registered Nurse Survey
57	Control over clinical practice	Alberta Registered Nurse Survey
58	Relationship between nurses and doctors [RN–MD relationships]	Alberta Registered Nurse Survey
59	Emotional exhaustion	Alberta Registered Nurse Survey
60	Flying status [condition of flight]	Safety beliefs and practices conducted by the Air and Surface Transport Nurses Association
61	Staff crew safety	Safety beliefs and practices conducted by the Air and Surface Transport Nurses Association
62	Patient safety	Safety beliefs and practices conducted by the Air and Surface Transport Nurses Association, SAQ, PSCHO
63	Scheduling and bed management	PMOS
64	Dignity and respect	PMOS
65	Core function support	PMOS
66	Burnout	OCSFS
67	Processes and equipment/resources	Patient Safety Culture Questionnaire in Acute Geriatric Units, Specific Questionnaire on Patient Safety in the Laboratory
68	Handoff	NHSOPSC
69	Nursing skills	ASCN
70	Psychological needs of patients	ASCN
71	Physical needs of patients	ASCN

Note: The instruments and their acronyms are described in the legend of Table 2.

## Data Availability

All data analyzed during this study are included in this published article.

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
