# Peer review of "Instruments for Patient Safety Assessment: A Scoping Review"

_healthcare, 2024, doi:10.3390/healthcare12202075_

Round 1
Reviewer 1 Report
Comments and Suggestions for Authors
The review is of significant interest to clinical risk managers, offering valuable insights into various tools for clinical risk assessment. However, to make it suitable for publication, I believe several clarifications regarding its purpose are necessary. It may be advisable to narrow the scope of the research by introducing more stringent eligibility criteria, focusing on specific dimensions explored by these tools (such as leadership, communication, teamwork, safety culture, stress, etc.) rather than covering all possible dimensions. In this regard, the title of the paper is perhaps too generic because the patient safety assessment can include also other tools not cited, for example, the investigation tools of adverse events.
While the selection of articles was methodologically sound, the review does not clearly convey the effectiveness and validity of the proposed tools. This could be addressed by refining the eligibility criteria.
The article selection indicates a focus on organizational and cultural factors (teamwork, communication, leadership, etc.) that influence the safety of care—a commendable objective. However, to truly assess the validity of these tools, outcome measures are essential. Indicators such as reductions in adverse events, healthcare-associated infections, mortality rates, and litigations would provide a clearer understanding of the tools' effectiveness. For example, the study on the WHO surgical checklist (N Engl J Med 2009;360:491-499) is particularly noteworthy as it evaluates outcomes (complications and mortality) based on the use or non-use of the checklist.
Another critical issue is the assessment of care safety based on the skills and competencies of healthcare professionals. A strong safety culture, effective communication, and teamwork are important, but these factors alone are insufficient without understanding how professionals (clinicians, surgeons, nurses) develop their competencies. In other words, the accreditation of organizational quality cannot be separated from the assessment of professional quality. The review should have included tools that evaluate this aspect, such as applying and analyzing learning curves to define proficiency levels. This often overlooked yet crucial aspect plays a significant role in ensuring the safety of care. The discussion should further explore the papers that address training.
Finally, as acknowledged by the authors, one of the review’s limitations is the quality of the included studies. By narrowing the research to certain dimensions, the focus could have been on higher-tier journals (Q1 and Q2), thereby emphasizing quality over quantity.
In the introduction, I would suggest two references for epidemiological studies that were cited. It is necessary to expand the introduction by better presenting and clarifying the tools currently used by clinical risk managers to carry out their work, including: incident analysis, organizational work analysis, the study of non-technical skills, and the introduction of best practices. The review will only examine some of these tools.
In the discussion, it would be useful to highlight how your review differs from others already published and which of these tools have led to a real improvement in patient safety. Lastly, patient safety depends not only on risk management organization but also on the skills and technical abilities of the professionals. Finally, it should be explained why the majority of these studies focus only on emergency services and not, for example, on other areas like surgery.
Reviewer 2 Report
Comments and Suggestions for Authors
The manuscript investigates through a well conducted scoping review the instruments for patient and culture safety assessment in healthcare.
The topic is consistent with Journal scope and the manuscript rigorously and briefly meets the aim of proposing the state on art on instruments and dimensions for patient and culture safety assessment in healthcare, opening for methods standardization and benchmarking.
I have only minor considerations, and the most important is proper referencing: please revise order as not answering appearance order also without link with the text (n. 15 refers to Italy not UK, n. 38 to not to Australia but only to Saudi Arabia ... and so on).
line 63 to 77 are the general indications of MDPI format to delete.
Figure 1: prisma for retrieval: probably missing a number.
Table 2: please add Nh and NF meanings in the legend
the spaces among words in table 3, Dimensions column.
Many thanks
Reviewer 3 Report
Comments and Suggestions for Authors
I ve read the paper "Instruments for Patient Safety Assessment: Review" I suggest to add in the title in the pdf scoping review.
The top
Abstract: I think that the abstract is sufficient.
I ve no remarks on introduction and methodology.
Discussion according to the chosen methodology is correctly done, however I think that this study has not real strength (those in lines 306 - 313 are the minimum standard for a scientific paper), on the contrary it doesn't offer a scenario based on the quality of the selected papers.
Round 2
Reviewer 1 Report
Comments and Suggestions for Authors
Thank you for your responses to my comments. In general, they address my remarks. I would only ask for a few clarifications, which I have noted in your cover letter.
Best regards

Reviewer 3 Report
Comments and Suggestions for Authors
Authors have made all the formal required correction.
Author Response
Reviewer 3:
Comments and Suggestions for Authors
Authors have made all the formal required correction.
Response: Thank you